Super Partition: fast, flexible, and interpretable large-scale data reduction in R

Queen Katelyn J. 1 kjqueen@usc.edu
Barrett Malcolm 2
http://orcid.org/0000-0001-7961-8943 Millstein Joshua 1
1 Department of Population and Public Health Sciences, University of Southern California , Los Angeles, California , United States
2 Department of Health Policy, Stanford University , Stanford, California , United States
Ferreira Pedro
Electronic publication date: 2025 Jan 27
Publication date: 2025
Volume: 13
Electronic Location ID: e18580
Received 2024 Apr 29; Accepted 2024 Nov 4
Copyright: © 2025 Queen et al.
Copyright year: 2025
Copyright holder: Queen et al.
License: This is an open access article distributed under the terms of the Creative Commons Attribution License, which permits unrestricted use, distribution, reproduction and adaptation in any medium and for any purpose provided that it is properly attributed. For attribution, the original author(s), title, publication source (PeerJ) and either DOI or URL of the article must be cited.
License URL: https://creativecommons.org/licenses/by/4.0/

Keywords: Data reduction, Clustering, Big data

Funding: National Institute of Environmental Health Sciences T32ES013678 National Heart Lung Blood Institute R01HL118455 National Institute on Aging P01AG055367 National Institute of Digestive and Kidney Disease R01DK110793 National Cancer Institute P01CA196569 This work was supported by the National Institute of Environmental Health Sciences (T32ES013678 to Katelyn J. Queen); the National Heart Lung Blood Institute (R01HL118455 to Joshua Millstein); the National Institute on Aging (P01AG055367 to Joshua Millstein); the National Institute of Digestive and Kidney Disease (R01DK110793 to Joshua Millstein); and the National Cancer Institute (P01CA196569 to Joshua Millstein). The funders had no role in study design, data collection and analysis, decision to publish, or preparation of the manuscript.

==============================
Motivation

As data sets increase in size and complexity with advancing technology, flexible and interpretable data reduction methods that quantify information preservation become increasingly important.

Results

Super Partition is a large-scale approximation of the original Partition data reduction algorithm that allows the user to flexibly specify the minimum amount of information captured for each input feature. In an initial step, Genie, a fast, hierarchical clustering algorithm, forms a super-partition, thereby increasing the computational tractability by allowing Partition to be applied to the subsets. Applications to high dimensional data sets show scalability to hundreds of thousands of features with reasonable computation times.

Availability and implementation

Super Partition is a new function within the partition R package, available on the CRAN repository (https://cran.r-project.org/web/packages/partition/index.html).

Background

Dimensionality reduction methods reduce complexity in data with the goal to reduce dataset dimensionality while preserving information and global relationships between features, leading to minimal information loss to allow for better data visualization and more efficient data processing. Partition is an agglomerative approach that divides data into sets of related features, where each set can be reduced to a single feature that captures a user-specified minimum amount of information. It’s a surjective approach in that each original feature maps to one and only one new feature, facilitating interpretability. The metric for information capture can be selected or designed by the user, e.g., mutual information for non-normal distributions or the intraclass correlation coefficient (ICC) for more normally distributed data. It’s also flexible in how features are reduced within a set, e.g., mean, first principal component, etc. Under some conditions likely to be common in real data, described in detail in Millstein et al. (2019), reduction with Partition can increase power to detect true associations as compared to principal component analysis (PCA), non-negative matrix factorization (NNMF), or no reduction. In comparison to algorithms like PCA, Partition preserves local data structure.

However, the original implementation is limited by memory and run-time, due to the requirement that a dissimilarity matrix be computed that includes all feature pairs. Thus, the maximum size dataset is restricted to tens of thousands of features, with exact size limitations dependent upon the user’s computing environment. In contrast, the Genie hierarchical clustering algorithm incorporates single linkage (Gagolewski, Bartoszuk & Cena, 2016), meaning that distances are computed between clusters rather than single features. Therefore, it is not subject to these constraints, allowing it to scale up to over one million features in reasonable run-time (Gagolewski, 2021). Although Genie was not designed as a data reduction method, it can be used to form super-partitions in a high-dimensional setting, allowing Partition to be applied to the components, yielding a computationally tractable reduction approach. We developed an approximation to Partition using Genie that scales the algorithm to big data, with run-time benchmarking for datasets up to almost one-half million features.

Software implementation

Portions of this text were previously published as part of a thesis (https://impa.usc.edu/asset-management/2A3BF1MGJ5LQF?FR_=1&W=1159&H=707).

In an initial step, Genie is used to form super-partitions of the data composed of ⌈Nc⌉ components or clusters, where N is the number of features in the full dataset, and c is the user-defined maximum cluster size (default value = 4,000). For initial super-partitions with size greater than c, Genie is applied in an iterative process to constrain cluster sizes to below c. Genie allows users to supply what they refer to as the Gini index, which is an economic inequity measure (range: [0, 1]). This measure aims to prevent formation of clusters with imbalanced sizes. We choose a threshold of 0.05 to highly penalize the formation of small clusters. However, for clusters of size Nk > c, small clusters may still form if features are highly correlated. Thus, if a large cluster is broken into smaller parts, and the smallest is of size 50 or less, we reject the Genie partition at this step and instead use k-means (MacQueen, 1967) with ⌈Nkc⌉ centroids to reduce the large cluster into more evenly sized parts. Finally, the Partition algorithm is applied to each super-partition cluster. A consequence of the super-partition step is that the minimum number of features in the final dataset is ⌈Nc⌉ or the initial number of super-partitions.

This process preserves the Partition capability of constraining information loss to a user-specified threshold on a local level. That is, information loss is constrained with respect to a limited number of related features within a cluster rather than the entire dataset. In contrast, if the user were to apply PCA and select the top principal components that explain some percentage of the variance of the data, information loss would be constrained on a global level.

Application

The computational efficiency of Super Partition was assessed and compared to Partition in several datasets, including transcriptome-wide microarray data in whole blood from the ABRIDGE and CAMP asthma cohorts (n = 865, number of features (f) = 19,428) (Covar et al., 2012; Torgerson et al., 2011), transcriptome-wide RNA-seq data in colon adenocarcinoma tumor tissue from The Cancer Genome Atlas (TCGA) (n = 481, f = 60,660) (Muzny et al., 2012), and Illumina 450K DNA methylation (DNAm) array data from the ABRIDGE cohort. The DNAm data were analyzed separately by chromosomes 1, 2, and 3 (n = 705, f = 98,216) and the full methylome (n = 705, f = 449,580).

CAMP and ABRIDGE expression profiles were measured via the Illumina Human HT-12 v4 array. All gene expression profile data were normalized via a log2-transformation and quantile-normalization within tissue. Expression profiles showed no major differences by sequencing date or clinical site, indicating an absence of technical batch effects. As previously described by the TCGA Network, TCGA expression profiles were measured via the Custom Agilent 244K Gene Expression Microarray (Muzny et al., 2012), and the data underwent extensive quality control checks, with minimal evidence of batch effects. The data were Loess normalized and log2 transformed. Finally, the DNAm data were measured on the Infinium HumanMethylation450 BeadChip array. Samples with poor intensity, unexpected beta value distribution, or with more than 5% of sites with a detection p-value greater than 0.05, were excluded. Normalization was performed on raw beta values using methods detailed in Liu & Siegmund (2016).

Partition and Super Partition were run 21 times each on the ABRIDGE whole blood gene expression data, using information loss criteria (the information loss criterion (ILC) threshold is the minimum ICC for any accepted cluster) in the range [0, 1] by 0.05 increments to allow for direct comparison of the methods. Changes in computation time and dimensionality reduction with varying maximum cluster sizes were measured by performing the Super Partition algorithm on the ABRIDGE whole blood gene expression data with an ILC of 0.10 and 0.60 at maximum cluster sizes of 250 and 500 to 4,500 in increments of 500. To test the scalability of the algorithm, Super Partition was applied to each of the three aforementioned datasets at the same 21 ILCs, all with a maximum cluster size of 4,000. Additionally, to test the large-scale capability of the algorithm, Super Partition was run across the full methylome (n = 705, f = 449,580) of the DNAm data again with a maximum cluster size of 4,000. All analyses were completed using the University of Southern California Center for Advanced Research Computing (CARC) cluster on a single Xeon-2640v3 2.60 GHz node with a single CPU and 16 GB of memory. Analyses were conducted using R version 4.2.3.

In a direct comparison to the base Partition algorithm across 19,428 features in the ABRIDGE whole blood gene expression dataset, Super Partition substantially outperformed Partition in computation time (Fig. 1A) across all thresholds. The time differences hits a maximum at an ILC of 0.35 and a minimum at the largest ILC values (ILC ∈[0.90,1]), where minimal dimension reduction occurs. The number of features in the reduced data for each algorithm are very similar for the entire range of correlation thresholds (Fig. 1B), showcasing that the approximation of the original Partition algorithm does not significantly alter the results. Using an ILC of 0.6 and a maximum cluster size of 4,000 (the function default values), Super Partition yields 15,348 features and Partition yields 15,253. Between the methods, 14,872, or 96.90%, of the features are the exact same, showcasing the similarity in the methods.

Figure 1 Comparing computation time and number of features in reduced data for partition and super partition.

Plot showing (A) time in hours and (B) number of clusters identified after partitioning (solid line) and super partitioning (dashed line) data across varying thresholds for minimum intraclass correlation coefficients in the ABRIDGE whole blood gene expression data (n = 865, number of features = 19,428). The default super partition maximum cluster size of 4,000 was used.

Figure 2 shows the time in minutes to Super Partition the ABRIDGE whole blood gene expression data at ILCs of 0.10 and 0.60 for varying maximum cluster sizes (function default value = 4,000). Reasonably, as the maximum cluster size increases and thus the complexity of each super-partition, so does the computation time for both ILCs. The number of features in the reduced data for these cluster sizes range from 949 to 652 or 95.12% reduction in the number of features to 96.67% for an ILC of 0.10, and 15,342 to 15,396 or 21.03% reduction in features to 20.75% for an ILC of 0.60.

Figure 2 Time in minutes to partition ABRIDGE whole blood gene expression with varying maximum cluster sizes.

Plot showing time in hours to complete super partition algorithm with an information loss criterion of 0.60 (red) and an information loss criterion of 0.10 (blue) for varying maximum cluster sizes in the ABRIDGE whole blood gene expression dataset. Percent reduction in features for the reduced data are displayed for selected maximum cluster sizes (250, 2,500, 4,500).

Figure 3A shows the time in hours to reduce the three datasets across the full range ([0, 1]) of ILCs. As expected, increasing numbers of features result in longer analysis times. Mid-range ILCs may be computationally expensive due to a large number of clusters and large sizes of clusters being formed, requiring more algorithm steps. Figure 3B shows the percent reduction in the number of features after using Super Partition across the range of ILCs.

Figure 3 Time and data reduction of super partition in varying datasets.

Plots showing (A) time in hours to complete the Partition algorithm and (B) percent reduction in features after partitioning data across varying thresholds for minimum intraclass correlation coefficients. ABRIDGE whole blood gene expression data is shown in green, TCGA scRNA-seq data in pink, and ABRIDGE whole blood 450 k DNA methylation in blue.

Additionally, the entire ABRIDGE 450k DNA methylation data (n = 705, number of features = 449,580) was partitioned with an ILC of 0.50, resulting in 62.75% reduction in features. This analysis took 4,374.40 min or just under 73 h to complete and required 45 GB of memory with a single CPU.

Conclusion

A limitation of the approach is that some sets of features may not be fully reduced. Feature reduction is only considered within each super-partition cluster, meaning that even if the information loss for combining super-partitions is below the user-specified information loss threshold, these clusters will not be combined. Additionally, given that current strategy to recombine results after applying Partition to each super-partition cluster requires sequential information, it is not feasible to run Super Partition in a multiprocessing environment. Future directions include a rework of this strategy to allow for parallel processing which could further reduce computational time.

The proposed solution approximates the original Partition method. That is, Genie is also an agglomerative algorithm, and it is solely used for an initial super-partition step, thus differences are likely to be modest. Information loss constraint and distance metric flexibility, feature mapping, and all other strengths and options of Partition are preserved in Super Partition. Application to real data shows that Super Partition greatly reduces computation time when directly compared to the original Partition algorithm and exponentially increases the number of features the algorithm is capable of handling.

We showed that maximum cluster size is inversely related to computation time, and as stated, there is a lower bound on the number of features which is dependent on maximum cluster size. Across the full range of maximum cluster sizes tested, there is a five-fold and eight-fold increase in computation times for ILCs of 0.60 and 0.10, respectively. However, the largest change in percent reduction in features occurs between a maximum cluster size of 250 and 2,500, and in this region, there is a 350% increase in computation time and less than a half-percent increase in percent reduction in features for an ILC of 0.60. Comparatively, for the analysis using an ILC of 0.10, there is also a 350% increase in computation time and a 27.29% increase in feature reduction. Therefore, we note the trade-off between dimension reduction and computation time, particularly for small ILCs. Users should consider analysis priorities when choosing these parameters.

The Super Partition algorithm is a fast, flexible way to reduce datasets while preserving information and interpretability of results, now feasible for datasets with hundreds of thousands of features.

Supplemental Information

Supplemental Information 1 Example code for running Super Partition.

This code includes a query to a portion of the data used for analysis as well as an example of how to complete the full analysis.

The authors would like to thank the study participants.

Additional Information and Declarations

Competing Interests

The authors declare that they have no competing interests.

Author Contributions

Katelyn J. Queen conceived and designed the experiments, performed the experiments, analyzed the data, prepared figures and/or tables, authored or reviewed drafts of the article, and approved the final draft.

Malcolm Barrett analyzed the data, authored or reviewed drafts of the article, and approved the final draft.

Joshua Millstein conceived and designed the experiments, authored or reviewed drafts of the article, and approved the final draft.

Data Availability

The following information was supplied regarding data availability:

The example code is available in the Supplemental File. The Super Partition algorithm code is available at GitHub and CRAN:

- https://github.com/USCbiostats/partition/blob/master/R/super_partition.R.

- 10.32614/CRAN.package.partition.

The third-party datasets are available at:

- GEO: GSE22324.

- TCGA: https://portal.gdc.cancer.gov/projects/TCGA-COAD.

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
