# Peer review of "Super Partition: fast, flexible, and interpretable large-scale data reduction in R"

_PeerJ, doi:10.7717/peerj.18580_

## Round 0.1 · original submission · Major Revisions

Due to differing opinions, the paper has been seen by five reviewers who have reviewed the manuscript in detail. Although there are some positive aspects highlighted by some of the reviewers, the opinions are quite divergent, with several critical points raised by reviewers 1 and 5 and reviewers 3 and 4.

In particular, the points regarding the clarity of the results, the significance of the experimental design and the results need to be adequately addressed before the paper can be considered for publication.

Reviewer 1 ·

Basic reporting

1. Content Clarity: The document lacks a clear structure, introduction, methodology, results, and discussion sections typically found in research articles. Adding these sections would enhance the clarity and organization of the content.

2. Research Contribution: The document does not present any research findings, analysis, or conclusions. Including a research question, hypothesis, methodology, results, and discussion would help demonstrate the scientific contribution of the work.

3. References and Citations: While the document includes references to various authors, it lacks context or explanation of how these references relate to the content. Providing proper citations and integrating them into the text would strengthen the credibility of the information presented.

Experimental design

1. Limited Information on Data Processing: The document does not elaborate on how the raw data was preprocessed, normalized, or cleaned before applying the Super Partition algorithm. Understanding these preprocessing steps is crucial for interpreting the results accurately.

2. Inadequate Statistical Analysis: The document lacks information on the statistical methods used to analyze the data and evaluate the performance of the algorithm. Without proper statistical analysis, it is challenging to determine the significance of the results obtained.

3. Scalability and Generalizability: While the document mentions scalability for datasets up to almost half a million features, it does not provide information on the generalizability of the algorithm to different types of datasets or domains. Assessing the algorithm's performance across various datasets would strengthen the experimental design.

Validity of the findings

1. Transparency and Clarity: The document lacks transparency in presenting the findings, as it does not provide detailed results or analysis procedures. Without clear and transparent reporting of the results, it is difficult to evaluate the validity of the findings.

2. Statistical Significance: The document does not mention statistical tests or measures of significance used to validate the findings. Without statistical analysis to support the results, the validity of the findings may be questionable.

3. Benchmarking and Comparison: While the document mentions comparisons between Super Partition and other methods, it does not provide detailed benchmarking results or comparisons with existing algorithms. Without robust benchmarking against established methods, the validity of the findings may be limited.

Reviewer 2 ·

Basic reporting

The article is well-written and easy to follow. Since this article is a follow-up to a previously published method (partition, published in Bioinformatics 2020), a brief background is provided. This is adequate given the nature of the work as a bioinformatics tool. The main purpose of this tool is to improve the speed of their previous method which they demonstrate using Figure 1 with speed-ups of greater than 100x using their new implementation (super partition). This work is self-contained in that its main focus is to improve the computational time of their previous method.

Experimental design

This article is based on a bioinformatic software tool which is within the aims and scope of PeerJ. The research question is well-defined and useful for the users of the software who may wish to apply this method to large datasets (hundreds of thousands of variables). It would be useful to compare how this method scales as the number of samples and features increase; the authors only apply this method to the ABRIDGE whole blood gene expression data only. Further applying this method to other types of data-types such as single-cell data will be useful to show that its application to datasets beyond bulk gene expression. The software is adequately described and the code works well through the partition R-package (provided on CRAN).

Validity of the findings

The authors used a high-performance cluster (HPC) to run the code. The example code provided also runs on my laptop when I reduce the number of features with no errors, however, I did not test this code on a HPC. The data used for this example code is based on a public GEO dataset (GSE22324) which I was able to download and analyse. The limitations and conclusions are clearly stated.

Reviewer 3 ·

Basic reporting

1. This article written in English is clear and unambiguous.
2. This article includes the background's introduction, and some relevant prior literatures are referenced.
3. Some figures appear in this article.

Experimental design

Although some experimental results are shown in the article, there are three problems as follows:
1. What effects does this super partition method have on the clustering external evaluation indicators such as accuracy, F-Measure and Purity?
2. When the super partition method is compared with other clustering algorithms based on data reduction or dimension reduction, which one is faster and more accurate?
3. What effects do the different parameters have on the super partition method? some related experimental results should be given.

Validity of the findings

no comment

Reviewer 4 ·

Basic reporting

I would first like to commend the authors for their work in the Super Partition function. Given the testing environment, the gains in computational time are impressive, leading to much shorter and almost linear runtimes in large molecular datasets irrespective of number of features as compared to the base Partition algorithm. What follows are my suggestions on how to improve readability and better explain the concepts in the paper, motivate the reader as to why and when they should use Super Partition over other dimensionality reduction methods, and suggestions for strengthening the results and conclusions.

Experimental design

In the Conclusion, there are two limitations to the experimental design that I would note:

1. The comparisons were not conducted in a multiprocessing environment, which is commonly available. To add to the importance of this limitation, Gagolewsky *et al.* (2021) mention that their Genie algorithm was built with multi-core systems in mind.

2. I would suggest adding information about future work where, for example, the authors may propose to assess performance between the Partition and Super Partition methods in the context of practical applications such as feature selection in the context of machine learning predictions.

Validity of the findings

In regards to the analysis of Figure 1 in lines 78-89, I do not see a visually compelling difference between the two distributions. Is there a significant difference as measured with e.g. p-value in the interval specified in line 79?

With respect to the analysis in the paragraph starting at line 80, were the authors able to run the Partition algorithm with the larger datasets described therein? If the authors can show that Super Partition can handle these larger datasets whereas Partition runs out of memory, that might be worth noting as an advantage of their algorithm.

In line 94, I'm unsure if we can fully know if Super Partition approximates the Partition method or not. The numbers of new features are similar, but can the authors qualitatively show how similar the results are?

Additional comments

Thorough the manuscript, I found that the concept of "super-partition" is not properly defined nor used consistently. For example, it is described as a singular entity comprised of several clusters in lines 46-47; meanwhile, the term "multi-partitions" is used in line 57, fostering the idea that multi-partition is a plural entity. Due to the importance of this concept, I would suggest defining it as clearly as possible and early in the manuscript (in particular, in the abstract, lines 17-18, and the Background), and use super-partition and related terms as consistently as possible.

In line 26, dimensionality reduction methods are incorrectly called "dimension reduction methods". Furthermore, I do not agree that the main goal is to eliminate noise while preserving signal, although it is one of the benefits from using these methods. Rather, I would say that the goal is to reduce dataset dimensionality while preserving the most important information and relationships from the original features. It might also be worthwhile to mention that the primary reason for applying these methods is to allow for more efficient data processing, visualization and analysis while minimizing information loss, to further underscore its importance.

In line 33, what are these conditions? How do the authors know these conditions are likely to be common in real data? I would suggest giving one or more illustrative examples.

Concerning lines 36-40, I would like to see the contrast drawn between Partition and Genie to be more incisive and concise. To that end, I would explain in simpler terms *why* Genie is not as constrained by memory limits and runtimes by emphasizing the computational differences between matrix computation of similarity across all feature pairs as opposed to computing similarity as the shortest distance between clusters of features.

There is some confusion in lines 49-51. The threshold in range \[0, 1] *is* the economic inequity measure, and should be defined as such. The authors should then mention that an example of these measures is the Gini index. I'm not quite sure what is meant by "supersede the threshold". Lastly, I would mention that the goal of the economic inequity measure, e.g. Gini index, relative to the Genie algorithm is to prevent the formation of clusters with highly imbalanced sizes.

To emphasize the importance of this research, I would have added a condensed version of lines 58-62 to the background, as there I was not properly convinced as to why I should use Partition over other methods such as PCA.

Smaller notes and corrections:
- For Figures 1 and 2, I would suggest adding an outline and/or background to the legends to make them more legible and eye-catching.
- Line 52: "However, **for** clusters size N_k > c..."
- Line 70: missing dimensions for the full methylome dataset.
- Line 72: mention amount of available RAM
- Lines 97-99: Partition or Super Partition? Because I would hesitate to call Partition fast, given the results!

Reviewer 5 ·

Basic reporting

1. The paper is generally well-written in clear and concise language, except for:
ILC should be defined at its first use, not its 7th. It is defined in the third paragraph of “Application”, but first used in the second.
2. The code available online through their github repository, https://github.com/USCbiostats/partition/blob/master/R/super_partition.R, is well-commented, and well-documented.
3. The code provided in the supplementary information works.
4. The tests in their software package appear inadequate; particularly, they only appear to check for very high-level and specific, and do not actually test for performant use. I tried to generate my own simulations to illustrate the algorithm's effectiveness (or lackthereof) in very basic simulation environments, as there aren't really performance benchmarks in the paper (see 2. and 3.). Very basic two-class prediction task example (n=200, dimensionality=101, mean-difference isotropic gaussian simulation, svm classifier with first 100 samples training, second 100 samples test). I found that this algorithm failed with very ambiguous errors frequently (frequently obtained "Error: Expecting a single value: [extent=0]." about 60% of the trials over 100 trials per attempt), while the others I compared it against did not yield errors. I tried benchmarking against prcomp, partition, and doing nothing at all; as I including partition(), which shares the same API, this suggests to me that I was indeed using it right; due to d = 101 < 4000, I set cluster_size << d typically between 5 and 20 somewhat arbitrarily. I tried for many combinations of hyperparameters for the cluster_size and always seemed to get high rates of failures. While the code is not the immediate contribution of the work, as there are no performance simulations to get a feel for how the algorithm actually performs I tried to make my own, and the algorithm did not run. Not sure if I did something wrong, but with this outcome I'd say the tests of the code are insufficient.

Experimental design

Similar comments to validity of the findings, in 3. While computational efficiency is important, there is a major tradeoff (is the result useful?) which has been (in my opinion) ignored in the context of this work. The problem and need is well-defined but I am not sure the applications demonstrate it.

Validity of the findings

1. A key feature of dimension reduction is that the “dimension-reduced” dataset preserves useful properties of the non-reduced dataset; presumably, with downstream applications in-mind. In the background in particular, the claim is made that: “Under some conditions likely to be common in real data, reduction with Partition can increase power to detect true associations”. The modifications of this technique relative partition seem to be minor; my understanding (and based on my read through of your code) is that you are basically splitting the dataset creatively, prior to serially applying partition serially; supported by your statement “increasing the computational tractability by allowing Partition to be applied to the subsets”. I think a majorly prohibitive limitation of the current work is that there is really no cohesive delineation that is similarly advantageous (downstream performance wise) as partition in the as-written paper, or other accelerated techniques for HDLSS data (e.g., PCA approaches designed for HDLSS data). This feels like a substantial limitation to me in that the claim of “thus differences are likely to be modest” is not well-supported, yet it is perhaps the most important claim of the entire paper to make.
2. The authors do not compare their technique to any best-practice approaches for dimensionality reduction in ultra-high sample size HDLSS data. The only comparison is to partition, which the motivation for this article implies is not well-suited for ultra-high sample size HDLSS data, and therefore is probably not a great comparison.

Additional comments

I think it would be in the best interest of the authors to very clearly delineate where this method fits in, and illustrate where it fits in, from a performance standpoint; as it stands, the take home of the paper feels more like "this is a procedure one could execute efficiently to big data", rather than "this is something one should do, and it happens to be efficient". I think this is a fundamental feature of dimensionality reduction unexplored in the paper nor its supplementary information.

---

## Round 0.2 · Minor Revisions

While two of the reviewers are satisfied with the current improved version of the manuscript, one of the reviewers has took the time to do additional testing of the software. Since he/she points failures in a significant number of times, it is important to properly address this issue so no major questions remain before a final decision is made. I believe that this can further improve the paper.

I add below the code that the reviewer used for testing:



require(partition)
require(MASS)
require(abind)
require(mclust)
require(e1071)
require(tidyverse)

n <- 200
n.train <- 100
d <- 101
R <- 20

effect_szs <- seq(0, 8, length.out=5)
Sigmas <- abind(diag(d), diag(d), along = 3)
pi <- 0.5

eval.dimred <- function(Xs, Ys, n.train) {
df.dr <- cbind(data.frame(Xs), data.frame("Class"=factor(Ys, levels=c("0", "1"))))
tr.svm <- svm(Class ~ ., data=df.dr[1:n.train,])

yhats <- predict(tr.svm, df.dr[(n.train + 1):n,])
return(mean(yhats == Ys[(n.train + 1):n]))
}

results <- do.call(rbind, lapply(effect_szs, function(sz) {
mus <- cbind(array(0, dim=d), sapply(1:d, function(i) sz/i^2))
do.call(rbind, lapply(1:R, function(i) {
Ys <- rbinom(n, 1, pi)
Xs <- t(sapply(Ys, function(y) mvrnorm(n=1, mu=mus[,y + 1], Sigma=Sigmas[,,y+1])))

Xs.part <- partition_scores(partition(Xs, threshold=.05))
# Inside the lapply loop, modify the tryCatch block:
acc.spart <- tryCatch({
Xs.spart <- partition_scores(super_partition(Xs, threshold=.05, cluster_size=15))
eval.dimred(Xs.spart, Ys, n.train)
}, error = function(e) {
print(e)
NA_real_ # Return NA of correct type
})
# arbitrarily select 25 dimensions
Xs.svd <- svd(Xs, nu=25)
Xs.pca <- Xs.svd$u %*% diag(Xs.svd$d[1:25])

# Then create the data frame only after all values are collected
acc.part <- eval.dimred(Xs.part, Ys, n.train)
acc.pca <- eval.dimred(Xs.pca, Ys, n.train)
acc.nothing <- eval.dimred(Xs, Ys, n.train)

# Ensure the data frame is created with all columns even when there's an error
return(data.frame(
Strategy=c("Partition", "Super Partition", "PCA", "Nothing"),
Accuracy=c(acc.part, acc.spart, acc.pca, acc.nothing),
i=rep(i, 4), # Repeat i for each row
Effect.Size=rep(sz, 4) # Repeat sz for each row
))
}))
}))

results %>%
group_by(Strategy) %>%
summarize(Count.NAs = sum(is.na(Accuracy)), Ntrials=n()) %>%
mutate(Rate.NAs = Count.NAs/Ntrials)

results %>%
group_by(Strategy, Effect.Size) %>%
summarise(SE=sd(Accuracy)/sqrt(n()), Accuracy=mean(Accuracy)) %>%
ggplot(aes(x=Effect.Size, y=Accuracy, color=Strategy)) +
geom_line()

Reviewer 3 ·

Basic reporting

no comment

Experimental design

no comment

Validity of the findings

no comment

Additional comments

Authors have already revised the manuscript according to comments. I think this manuscript can be accepted.

Reviewer 4 ·

Basic reporting

The authors have successfully addressed my concerns regarding the article.

Experimental design

The authors have successfully addressed my concerns regarding the article.

Validity of the findings

The authors have successfully addressed my concerns regarding the article.

Additional comments

The authors have successfully addressed my concerns regarding the article.

Reviewer 5 ·

Basic reporting

Thank you for your previous responses. I wanted to follow up regarding some ongoing technical challenges I'm experiencing with the code. I've updated to the latest version from GitHub (~3 weeks ago) - and while I had previously installed the updated version that supposedly fixed the reported error, I reinstalled just in case. However, the error persists.

I understand and appreciate your responses to the experimental design questions and validity findings. I note that my additional comments were somewhat "glossed over," with explanations provided more in response to Reviewer 1's comments, specifically: "We appreciate this point, however, given that this manuscript is meant to be a short software article, we believe that this is beyond the scope." While I respect this scope limitation, I believe the current technical issues with the software implementation need to be addressed for the article to fulfill its intended purpose.

I've been testing with a straightforward simulation involving a mean-difference between 2-classes and an SVM classifier on dimension-reduced data. While your "partition" function works perfectly with identical input arguments, the "super partition" function generates errors approximately 50% of the time. The error messages are quite non-specific, and it's worth noting that every other algorithm functions elegantly and performantly on this example. Additionally, I find the test cases included in the software package insufficient for code that hasn't undergone formal simulations and benchmarking.

Given that partition and super partition are described as functionally similar in your explanations (and therefore new simulations were deemed unwarranted), they should perform consistently with the same inputs. If there are implementation differences causing these divergent behaviors, it would be valuable to either understand the specific conditions causing these errors or consider more thorough simulations to validate the implementation.

I've attached my code that demonstrates these issues - showing failures ~50% of the time for super partition, while all other techniques work without error. It remains possible this is a machine-specific issue for me. If that's the case, figuring out why would be an reasonable solution, or a docker container made publicly available that is able to implement these basic test-cases or something of the like since I am quite computer/terminal/command-line savvy and have a lot of experience with R, C++, and RCpp, and was unable to decipher in ~1 hour of tinkering with the code this error only arises sporadically so I anticipate others may run into it as well.

Experimental design

I am content with my comments about experimental design are sufficiently addressed.

Validity of the findings

I am content with my comments about validity are sufficiently addressed.

Additional comments

I want to preface this with, I think this paper is awesome, and much more awesome with the revisions addressed to me and the other reviewers, and has some great potential. I am really not intending to come off as rude/stubborn in my comments about the lack of simulations/test cases for the code and the seemingly random errors that arise; I simply think for your work to have the most impact, having one (or the other) ironed out, and a more cohesive understanding of error-cases when the functions are applied to real data (since in your words, this is a short software article), will really be beneficial for you guys. So please take my comments with the grain of salt that, I want to be helpful for you as a test-dummy for your package/software so it can have good reach :)

---

## Round 0.3 · accepted · Accept

The authors have addressed the reviewers comment regarding the software issue detected by the reviewer.